# MMP9 modulates the metastatic cascade and immune landscape for breast cancer anti-metastatic therapy

Mark Owyong[1,*], Jonathan Chou[1,2,5,*], Renske JE van den Bijgaart[1], Niwen Kong[1], Gizem Efe[1], Carrie Maynard[1], Dalit Talmi-Frank[6], Inna Solomonov[6], Charlotte Koopman[1], Elin Hadler-Olsen[1], Mark Headley[3], Charlene Lin[1], Chih-Yang Wang[1], Irit Sagi[6], Zena Werb[1,5], Vicki Plaks[1,4]

**Metastasis, the main cause of cancer-related death, has traditionally been viewed as a late-occurring process during cancer progression. Using the MMTV-PyMT luminal B breast cancer model, we demonstrate that the lung metastatic niche is established early during tumorigenesis. We found that matrix metalloproteinase 9 (MMP9) is an important component of the metastatic niche early in tumorigenesis and promotes circulating tumor cells to colonize the lungs. Blocking active MMP9, using a monoclonal antibody specific to the active form of gelatinases, inhibited endogenous and experimental lung metastases in the MMTV-PyMT model. Mechanistically, inhibiting MMP9 attenuated migration, invasion, and colony formation and promoted CD8[+] T cell infiltration and activation. Interestingly, primary tumor burden was unaffected, suggesting that inhibiting active MMP9 is primarily effective during the early metastatic cascade. These findings suggest that the early metastatic circuit can be disrupted by inhibiting active MMP9 and warrant further studies of MMP9-targeted anti-metastatic breast cancer therapy.**

## Introduction

Most cancer-related deaths are due to metastatic disease. Metastasis, one of the classic "hallmarks of cancer" (Hanahan & Weinberg, 2011), is a multistage process that includes remodeling the local tumor microenvironment (TME), followed by invasion of tumor cells into the blood or lymph, survival in circulation, extravasation, and growth in a new microenvironment. The recognition that cancer is a systemic disease has been illustrated by studies showing the importance of various cell types in creating a metastatic niche (Lambert et al, 2017), and the role of the immune system in tumor growth (Aguado et al, 2017). However, although numerous studies have delineated mechanisms during the late stages of metastasis, there is little understanding about how early these niches are initiated during tumorigenesis and how they can be disrupted from a therapeutic standpoint. Importantly, there are currently no approved therapies that specifically aim to inhibit new sites of tumor growth.

The ECM, a critical component of the TME, undergoes extensive remodeling during breast cancer (BC) evolution. Matrix metalloproteinases (MMPs), a family of zinc-dependent endopeptidases, are pivotal players in ECM remodeling during cancer initiation and progression via multiple mechanisms (Kessenbrock et al, 2010; Bonnans et al, 2014). For example, in the primary tumor, MMPs cleave, degrade, and rearrange the components of the ECM. In addition, MMPs activate cytokines and release sequestered growth factors, thereby regulating many different pathological processes (Noel et al, 2012). Notably, active (rather than total) levels of circulating proteases, including MMP2 and MMP9, play a role in human BC classification and progression (Somiari et al, 2006). MMP9 expression correlates with more aggressive subtypes of BC and is associated with a higher incidence of metastasis and relapse (Vizoso et al, 2007; Waldron et al, 2012; Yousef et al, 2014). Furthermore, MMP9 is instrumental in establishing the metastatic niche (Hiratsuka et al, 2002; Kaplan et al, 2005) and functions as a key mediator in metastatic progression.

In the metastatic niche, myeloid cells have been implicated in locally supplying the niche with abundant quantities of MMPs (Yan et al, 2010). We and others have shown that CD11b[+]Gr1[+] myeloid cells accumulate in the metastatic lungs of mammary tumor virus (MMTV) promoter-driven polyoma middle T antigen (PyMT) mice (Kowanetz et al, 2010; Casbon et al, 2015; Wculek & Malanchi, 2015) and are involved in mediating metastatic niche formation. These myeloid cells decrease IFNγ production and elevate T helper 2 (Th2) cytokines, while producing large quantities of MMP9 that later promote vascular remodeling (Yan et al, 2010). This suggests that these cells alter the overall landscape of the metastatic lung microenvironment and tip the balance of immune protection to tumor promotion. Although immune subsets play the predominant role in supplying MMPs, tumor cells also contribute a variety of MMPs that aid in priming the metastatic niche (Köhrmann et al, 2009).

[1]Department of Anatomy, University of California, San Francisco, CA, USA   [2]Department of Medicine, University of California, San Francisco, CA, USA   [3]Department of Pathology, University of California, San Francisco, CA, USA   [4]Department of Orofacial Sciences, University of California, San Francisco, CA, USA   [5]The Helen Diller Family Comprehensive Cancer Center, University of California, San Francisco, CA, USA   [6]Department of Biological Regulation, Weizmann Institute of Science, Rehovot, Israel

Correspondence: zena.werb@ucsf.edu; vicki.plaks@ucsf.edu
*Mark Owyong and Jonathan Chou contributed equally to this work

Not surprisingly, given the pleiotropic roles of MMPs in cancer transformation and metastasis, there has been considerable clinical interest in MMP inhibitors. However, most clinical trials using small molecule MMP inhibitors were unsuccessful because of poor tolerability and lack of efficacy stemming from low specificity (Coussens et al, 2002). In addition, MMP inhibitors were given as monotherapy in patients with advanced metastatic disease and, thus, the therapeutic potential of targeting MMPs at earlier stages of disease was not evaluated (Deryugina & Quigley, 2006). By mimicking how endogenous tissue inhibitors of metalloproteinases bind to MMPs, we generated a specific blocking monoclonal antibody (referred to as SDS3) directed against the catalytic zinc–protein complex and enzyme surface (exosites) of the activated forms of MMP2 and MMP9 (Sela-Passwell et al, 2011).

Because metastatic dissemination is initiated early during tumorigenesis, we used the MMTV-PyMT immune competent mouse model. This model recapitulates the natural progression of human luminal B BC, which is hormone receptor–positive (estrogen receptor and/or progesterone receptor–positive) and either HER2 positive or negative and exhibits a less favorable prognosis compared with luminal A BC (Inic et al, 2014). In MMTV-PyMT mice, the expression of biomarkers in tumors is similar to those associated with poor outcomes in humans, including the loss of estrogen receptor and progesterone receptor and the persistent expression of ErbB2/Neu and cyclin D1 (Lin et al, 2003). In this study, we show that in MMTV-PyMT mice, a metastatic niche is established early during tumorigenesis and promotes lung colonization by circulating tumor cells (CTCs). Using complementary genetic and pharmacological approaches, we demonstrate that MMP9 plays a critical role in these early steps of metastasis. Blocking active MMP9 early during tumorigenesis inhibits metastatic colonization and opens a strategy to effectively target metastatic disease.

# Results

### A lung metastatic niche is established early during BC transformation

To investigate the lung metastatic niche in early BC, we compared the ability of i.v.–injected tumor cells to grow in the lungs of female MMTV-PyMT mice and their wild-type (WT) littermates. We used an MMTV-PyMT–derived cell line generated by Lynch and colleagues (Halpern et al, 2006) that we labeled with GFP and luciferase (referred to here as probing VO-PyMT cells) to mimic CTCs and probe their ability to colonize the lungs. Reporter genes in VO-PyMT cells distinguished them from any endogenous, spontaneous metastases. To understand the changes early during tumorigenesis, we i.v. injected the probing VO-PyMT cells into 6-wk-old MMTV-PyMT and WT littermates (Fig 1A). At this stage, MMTV-PyMT mice begin to form hyperplasias and early adenomas, and, importantly, there were no spontaneous micrometastatic foci present in the lungs (Fig S1A). We then monitored the mice longitudinally using bioluminescent imaging. 4 wk after i.v. injection, MMTV-PyMT mice showed a significantly increased lung metastatic burden from the injected probing cells compared with their WT littermates (Figs 1B and S1B). Interestingly, 2 wk after i.v. injection, when

the bioluminescent signal was still undetectable, MMTV-PyMT mice already exhibited a higher micrometastatic burden than WT mice by microscopic examination of the GFP signal (Figs 1C and S1C). Although PyMT is not expressed on the cell surface, cross-presentation of endogenous antigens could hypothetically lead to immune tolerance in MMTV-PyMT and not WT mice. To mitigate the concern that VO-PyMT cells colonize the lungs better in MMTV-PyMT mice because of immune tolerance to the PyMT antigen expressed in the primary tumor (Lin et al, 2003), we injected an independent, non-PyMT lung adenocarcinoma cell line (LAP0297, derived from FVB background [Huang et al, 2008]) into MMTV-PyMT and WT littermates (Fig 1D and E) and also observed increased lung metastatic burden in MMTV-PyMT mice. These data demonstrate that the lung microenvironment in MMTV-PyMT mice is more permissive to the growth of i.v.–injected probing cells, even during the pre-metastatic stages of tumorigenesis.

To further support our observations that systemic alterations occur early during tumorigenesis, we used a non-metastatic MMTV-PyMT model (DB-MMTV-PyMT), which carries a mutation in PyMT that decouples it from phosphatidylinositol 3-kinase (PI3K) signaling. These mice form hyperplasias and adenomas in all mammary glands, but rarely progress to carcinoma or develop lung metastases (Hutchinson et al, 2001). We found that DB-MMTV-PyMT mice i.v. injected with VO-PyMT cells were also more permissive for metastatic growth when compared with WT littermates (Fig S1D and E). This further suggests that early events during tumor progression promote metastatic colonization and that factors from the hyperplastic primary tumor prime the lung microenvironment.

### Systemic inhibition of active MMP9 abrogates metastatic growth in the lungs of MMTV-PyMT mice but does not attenuate primary tumor growth

To understand the changes that occur in these early hyperplasias, we profiled gene expression and found that at 6 wk of age, MMTV-PyMT hyperplasias expressed increased levels of multiple ECM components, remodeling enzymes (including MMPs), angiogenic and inflammatory factors, and Th2 cytokines implicated in mediating metastatic niche formation (Fig S1F) (Hiratsuka et al, 2002; Yan et al, 2010; Casbon et al, 2015).

Moreover, previous work demonstrated that inflammatory myeloid cells are a major source of MMP9 in primary tumors, and that these cells are critical in establishing the lung metastatic niche (Kowanetz et al, 2010; Yan et al, 2010). We sought to characterize this population of neutrophils and monocytes, which expresses the surface markers CD11b and Gr1. Although cells expressing these markers are frequently called myeloid-derived suppressor cells (Markowitz et al, 2013), in the MMTV-PyMT model most cells closely resemble neutrophils (Casbon et al, 2015). These CD11b$^+$Gr1$^+$ cells participate in ECM degradation, angiogenesis, and suppression of CD4$^+$ and CD8$^+$ T cell proliferation (Yan et al, 2010; Casbon et al, 2015). 6-wk-old MMTV-PyMT mice i.v. injected with VO-PyMT cells accumulated CD11b$^+$Gr1$^+$ cells as well as CD11b$^+$ DCs, macrophages, and conventional monocytes (migratory DCs that activate T cells) in the lungs after a 3-wk chase, as compared with WT littermates (Figs 2A and S2A). The CD11b$^+$Gr1$^+$ cells formed clusters surrounding metastatic islands of GFP$^+$ VO-PyMT cells, whereas Gr1$^+$ cells were sparse but evenly distributed within GFP$^-$ areas of the lungs (Fig 2B).

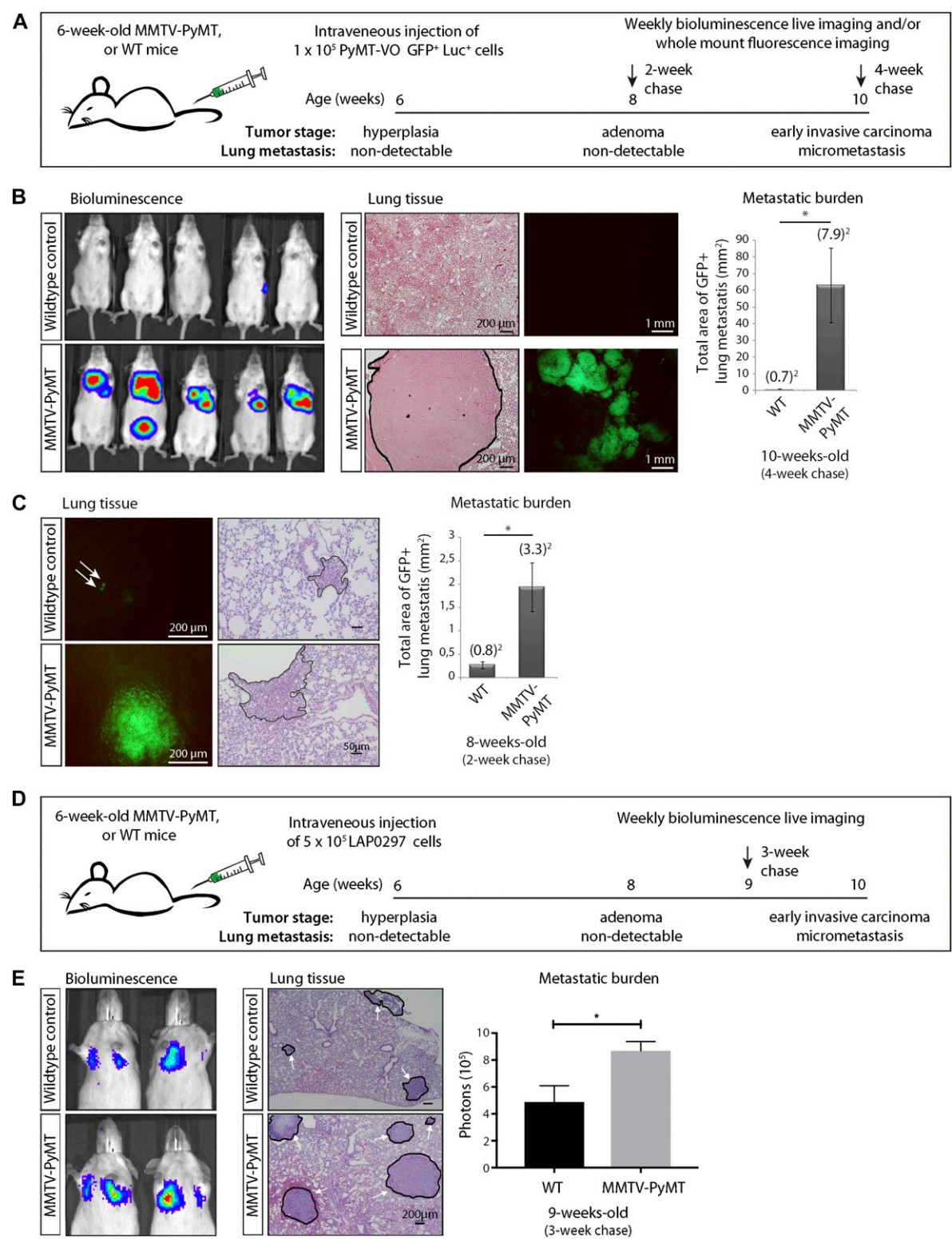

**Figure 1. Characterization of a pre-metastatic niche in the lungs of MMTV-PyMT mice.**
The lung microenvironment of hyperplastic MMTV-PyMT mice is permissive to the growth of i.v. injected circulating VO-PyMT tumor cells. **(A)** Schematic of the experimental setup: 6-wk-old MMTV-PyMT mice or WT littermate controls were i.v. injected with $1 \times 10^5$ VO-PyMT-GFP-Luc cells. The mice were monitored weekly using bioluminescence imaging and euthanized 2 and 4 wk after i.v. injection. The lungs were evaluated by whole-mount fluorescence imaging to detect and quantify lung metastases. **(B)** Left: bioluminescence imaging of WT littermate control (top) and MMTV-PyMT (bottom) mice 4 wk after i.v. injection of VO-PyMT-Luc-GFP probing cells (n = 10 WT and n = 10 PyMT). Middle: left—representative hematoxylin and eosin (H&E)–stained images of lung metastases in WT littermate control and MMTV-PyMT mice. Scale bar is 200 $\mu$m;

We next examined MMP2 and MMP9, which are expressed in primary MMTV-PyMT tumors (Fig S2B), across BC subtypes of human primary tumors and metastases (including nine matched pairs) to the chest wall, lymph nodes, lungs, liver, and spleen (Waldron et al, 2012). Among all 24 MMPs examined using publicly available microarray data (Waldron et al, 2012), MMP9, which is known to be pro-tumorigenic, was the only one significantly elevated in metastasis (Figs 2C and S2C). After i.v. injection of VO-PyMT cells, MMP9 co-localized with Gr1$^+$ cells but was also detected in Gr1$^-$ cells in the lungs of MMTV-PyMT mice (Fig 2D). This suggests that these myeloid cells are a predominant cell type that secretes MMP9 into the MMTV-PyMT metastatic niche, although other cells, including myeloid, stromal, and/or tumor cells, may also be a source of MMP9 (Psaila & Lyden, 2009) and contribute to metastatic colonization. We then examined the functional relevance of MMP9 by using MMP9 KO mice (Vu et al, 1998) to assess the effect of genetic ablation of MMP9 in the MMTV-PyMT model. Using MMTV-PyMT; MMP9 KO mice i.v.-injected with VO-PyMT cells, we found that MMP9 KO mice have reduced lung metastasis with no change in primary tumor burden (Fig 2E and F). This provided a proof-of-concept for the potential importance of MMP9 in metastatic growth and underlied MMP9 as a potential therapeutic target within the TME.

Using *in situ* zymography, we found that the lungs of MMTV-PyMT mice had significantly higher levels of gelatinase activity at adenoma and carcinoma stages of tumorigenesis, starting at 8 wk of age as compared with WT controls (Figs 2G and S2D). Interestingly, the increase in gelatinase activity coincided with the time point at which we detected a significant difference in the ability of VO-PyMT cells to colonize the lungs in the MMTV-PyMT model (Fig 1C). Collectively, these data suggest that increased MMP9 expression and activity in the lungs may be an important mediator in initiating and maintaining the metastatic niche.

Since active MMP9 is more relevant to disease than total MMP9 (Somiari et al, 2006), we next determined whether blocking active MMP9 would alter the ability of VO-PyMT cells to colonize the lungs of MMTV-PyMT mice. We treated MMTV-PyMT mice with SDS3 or control immunoglobulin (IgG), starting before and continuing every 2 d after i.v. injection of the VO-PyMT cells (Fig 2H). Although both MMP2 and MMP9 were present in primary tumors (Fig S2B), SDS3 reduced the metastatic growth of VO-PyMT probing cells in the lungs of MMTV-PyMT mice (Fig 2I) without affecting primary tumor growth (Fig 2J). We verified that SDS3 treatment reduced gelatinase activity in the lungs using in situ zymography (Figs 2K and S2E). These results demonstrated that metastatic colonization is facilitated by active MMP9 and that blockade of its activity primarily suppressed metastatic tumor growth in the lungs, without affecting primary tumor burden.

## SDS3 is biodistributed to tumor and metastatic foci and specifically associated with myeloid cells

Because SDS3 treatment *in vivo* attenuates metastatic colonization in MMTV-PyMT mice, we examined the biodistribution of SDS3. We crossbred MMTV-PyMT mice with the ACTB-ECFP (*β-actin* promoter driving enhanced cyan fluorescent protein) and c-fms (colony stimulating factor 1 receptor)-EGFP reporter mice to enable visualization of different tumor stages and myeloid cell dynamics. Using fluorescently-labeled SDS3 (SDS3-HyLite 555, i.p. injected 3 or 24 h before imaging) in the transgenic reporter models, we observed SDS3-HyLite 555-positive cells within the stroma surrounding the primary tumor foci (Figs 3A and S3A, Videos 1, 2, and 7). Some cells that took up SDS3-HyLite 555 also took up dextran, suggesting that these are macrophages expressing CD68 and CD206 (Egeblad et al, 2008), with some nonspecific uptake because of opsonization or through Fc receptors on immune cells (Fig S3B, Video 8). Within the sites of lung metastasis, we observed a marked increase in c-fms-EGFP, dextran, and SDS3-HyLite 555 triple-positive macrophages surrounding lung metastatic foci (Figs 3A and S3C). These results demonstrate that SDS3 labels myeloid cells that have invaded into the TME, possibly because of a high concentration of active MMP9 around these cells.

To understand the dynamics of SDS3 within the lung microenvironment further, we performed ex vivo time-lapse imaging of the lungs of MMTV-PyMT mice. We observed that SDS3-HyLite 555+ cells swarm around GFP$^+$ metastatic foci (Figs 3B and S3D, Videos 3 and 4). Flow cytometric analysis revealed that SDS3-HyLite 555 was taken up by a number of different myeloid cell types, including macrophages as well as patrolling and conventional monocytes (Fig 3C).

To confirm that SDS3 accumulates in tumor-bearing organs, we conjugated SDS3 to a far-red fluorophore (SDS3-HyLite 750), which visualized the biodistribution using bioluminescent whole-body imaging as well as *ex vivo* fluorescent and bioluminescent whole-mount imaging. In late-stage tumor-bearing MMTV-PyMT mice, the antibody accumulated in the primary tumor (Fig 3D). Similarly, in WT mice i.v.-injected with VO-PyMT cells, SDS3-HyLite 750 accumulated in the metastasis-bearing lungs, as well as in the liver, kidneys, and spleen, organs that clear antibodies (Fig 3D). SDS3-HyLite 750 was retained in the metastatic lungs 24 h postinjection, when IgG-HyLite 750 had cleared (Fig 3E). These results indicate that SDS3 accumulates within both primary and metastatic tumor foci.

## Inhibition of MMP9 activity by SDS3 attenuates migration, invasion, and colony formation of metastatic cells

Gene expression profiling by quantitative PCR (qPCR) showed that VO-PyMT cells express higher levels of *Mmp9* transcripts and higher

right—representative fluorescent whole lung images of WT control and MMTV-PyMT mice. Right: metastatic burden as quantified by the GFP$^+$ signal in WT control and MMTV-PyMT mice (n = 13 WT and n = 15 PyMT, P = 0.032) 4 wk after i.v. injection of VO-PyMT-GFP-Luc cells. **(C)** Left: representative fluorescent whole lung images after a 2-wk chase in WT control and MMTV-PyMT mice. Scale bar is 200 $\mu$m. White arrows point to GFP$^+$ metastatic foci. Middle: representative H&E images of lung metastases in WT littermate control and MMTV-PyMT mice. Right: quantification of the metastatic burden after a 2-wk chase in WT control and MMTV-PyMT mice (n = 11 WT and n = 12 PyMT, P = 0.014). **(D)** Schematic of the experimental setup: 6-wk-old MMTV-PyMT mice or WT littermate controls were i.v. injected with 5 × 10$^5$ LAP0297-GFP-Luc cells. Mice were monitored weekly using bioluminescence imaging and euthanized 3 wk after i.v. injection. The lungs were evaluated by whole-mount fluorescence imaging to detect and quantify lung metastases. **(E)** Left: bioluminescence imaging of WT control (top) and MMTV-PyMT (bottom) mice after i.v. injection with 5 × 10$^5$ LAP0297-GFP-Luc probing cells. Middle: representative H&E images of lung metastases in WT control and MMTV-PyMT mice after a 3-wk chase. Scale bar is 200 $\mu$m. White arrows point to regions of metastatic foci. Right: quantification of bioluminescence signal in WT control and MMTV-PyMT mice i.v. injected with LAP0297-GFP-Luc cells (n = 4 WT and n = 4 PyMT).

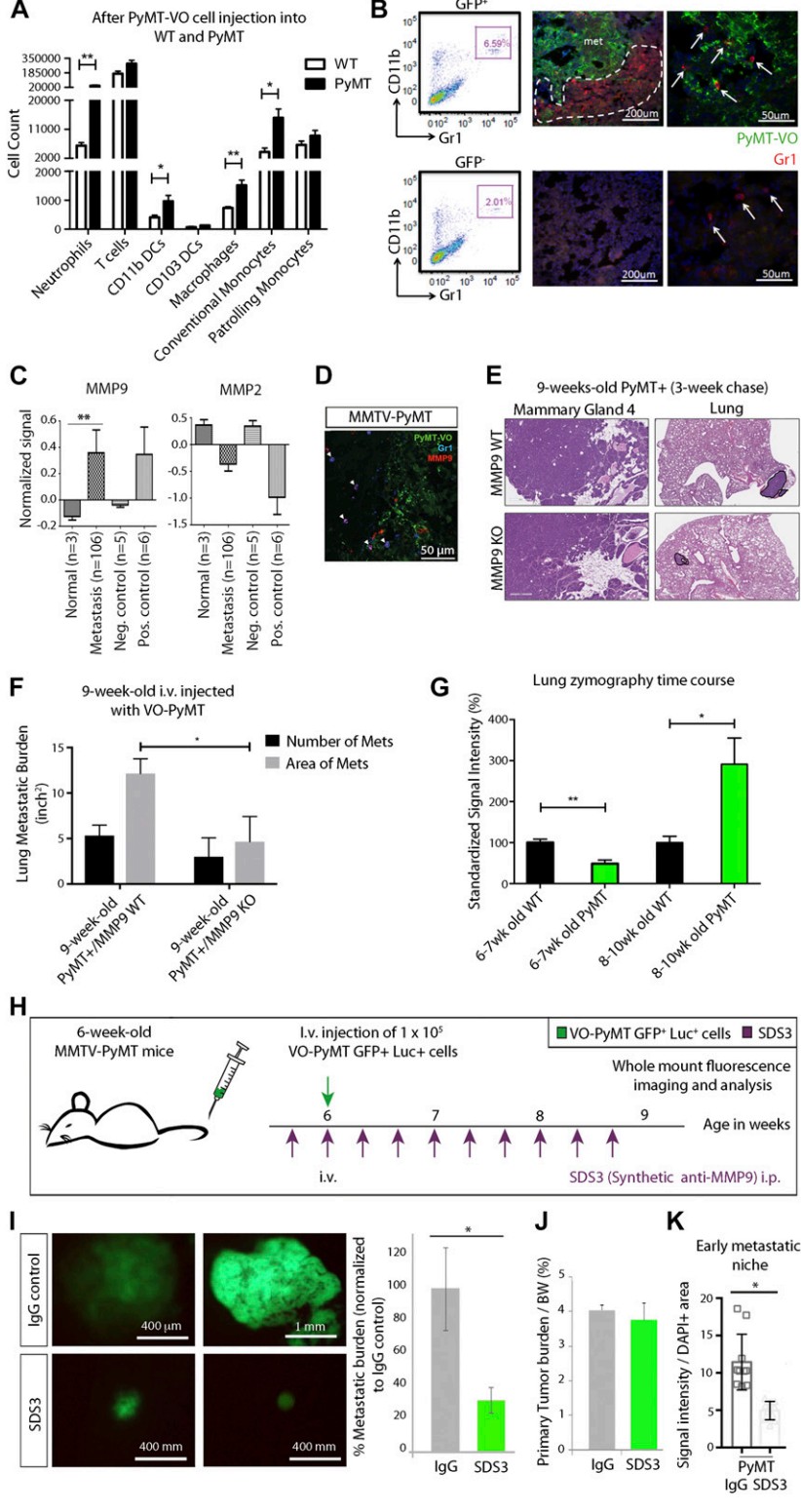

**Figure 2. MMP9 is present in cells within the metastatic microenvironment.**
Early niche formation is accompanied by increased inflammatory CD11b⁺Gr1⁺ myeloid cells that surround islands of metastatic cells in the lung. **(A)** Flow cytometric quantification of neutrophils (Ly6G⁺CD11b⁺), T Cells (CD3⁺CD11b⁻), CD11b DCs (CD11b⁺CD103⁻), CD103 DCs (CD103⁺CD11b⁻), macrophages (CD11c⁺Ly6C⁺Ly6G⁻), conventional monocytes (Ly6C⁺CD11c⁻), and patrolling monocytes (CD11c⁺Ly6C^low) gated on live CD45⁺ in the lungs of 9-wk-old WT littermates and MMTV-PyMT females 3 wks after i.v. injection of VO-PyMT-GFP-Luc cells (n = 4 WT, n = 3 MMTV-PyMT). **(B)** Left: flow cytometry analysis of CD11b⁺Gr1⁺ cells in GFP⁺ and GFP⁻ areas of MMTV-PyMT 2 wk after i.v. injection of VO-PyMT-GFP-Luc cells. Middle and right: confocal microscopy of 8-wk-old MMTV-PyMT lungs stained with GFP (green) to mark the injected VO-PyMT-GFP-Luc cells and Gr1 (red). Nuclei are stained with DAPI. Gr1+ (arrows) cells accumulate around the metastatic GFP⁺ foci but not in GFP⁻ areas. **(C)** Bioinformatics analysis of 90 BC patients with nine matched pair samples of chest wall, lymph node, lung, liver, and spleen (Waldron et al, 2012). MMP9 is significantly increased (P = 0.0014) across various metastatic organs, whereas MMP2 is decreased across various metastatic organs compared with primary breast tumors. **(D)** Confocal microscopy of MMTV-PyMT lungs stained with GFP (green), MMP9 (red), and Gr1 (blue) shows colocalization of MMP with Gr1+ cells (denoted by arrowheads). **(E)** VO-PyMT-GFP-Luc cells were i.v. injected into 6-wk-old MMTV-PyMT; Mmp9 WT or MMTV-PyMT; Mmp9 KO. H&E images of the primary tumor and lung in Mmp9 WT and Mmp9 KO mice after a 3-wk chase. **(F)** Quantification of the lung metastatic burden in MMTV-PyMT; Mmp9 WT and MMTV-PyMT; Mmp9 KO mice after i.v. injection of VO-PyMT-GFP-Luc cells (n = 4 MMTV-PyMT+; Mmp9 WT, n = 3 MMTV-PyMT+; Mmp9 KO). Mammary gland scale bar is 300 μm and lung scale bar is 700 μm. **(G)** Quantification of in situ zymography in lungs of WT control and MMTV-PyMT mice at various ages encompassing the hyperplasia to carcinoma transition (6–7-wk old: n = 6 WT, n = 7 MMTV-PyMT; 8–10-wk old: n = 4 WT, n = 4 MMTV-PyMT). Four serial sections stained. **(H)** Schematic of experimental setup: SDS3 or IgG (isotype control) is preinjected i.p. into 6-wk-old MMTV-PyMT mice. 1 d later, 1 × 10⁵ VO-PyMT-GFP-Luc cells, along with SDS3 or IgG, are i.v. injected into the mice. Subsequently, SDS3 or IgG is i.p. injected every other day for eight additional injections. **(I)** Left: whole-mount lung fluorescence imaging of lung metastases in MMTV-PyMT mice treated with SDS or IgG. Right: quantification of metastatic burden in MMTV-PyMT mice treated with SDS or IgG. (n = 8 IgG and n = 9 SDS3; *P = 0.02). **(J)** Quantification of primary tumor burden/body weight in MMTV-PyMT mice treated with SDS or IgG (n = 8 IgG and n = 9 SDS3, P = not significant). **(K)** Quantification of in situ zymography of the lungs in SDS3 and IgG-treated MMTV-PyMT mice (n = 3 each).

levels of active MMP9 compared with non-transformed mouse mammary epithelial cells (Figs 4A and S4A). To determine the effects of SDS3 directly on metastatic BC cells, we first i.v. injected ten times more VO-PyMT cells into 6-wk-old WT mice compared with

MMTV-PyMT mice (Fig 4B). This would saturate the system with probing cells so that these would potentially supply enough MMP9 locally to facilitate their colonization. Mice treated with the SDS3 antibody had significantly reduced metastatic growth and reduced

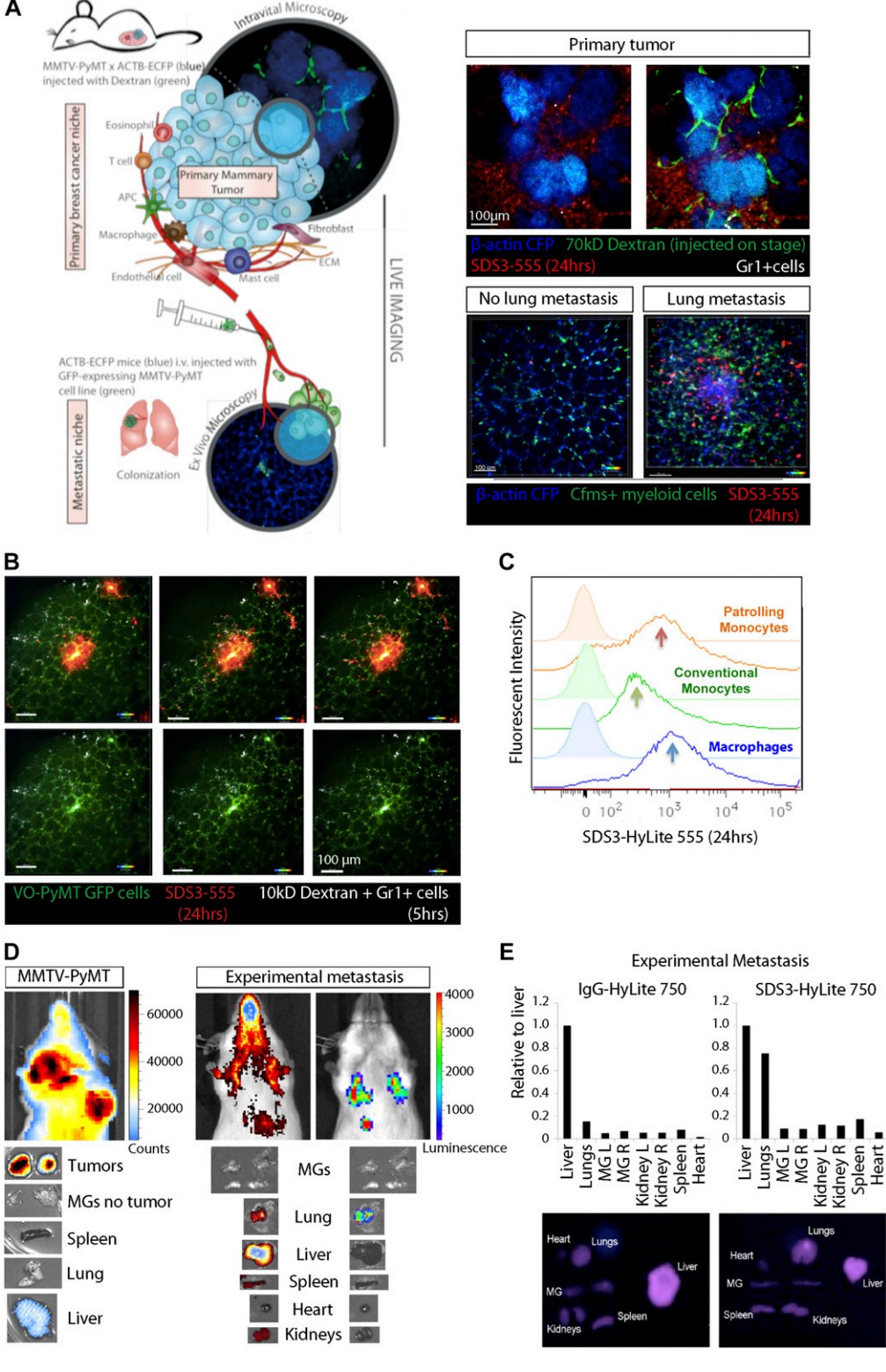

**Figure 3. SDS3 concentrated within and around primary tumor and metastatic foci.**

**(A)** Left: schematic of experimental setup for imaging MMTV-PyMT; ACTB-ECFP i.v. injected with VO-PyMT-GFP-Luc cells and SDS3-HyLite 750. Right: top—primary tumor confocal microscopy 24 h after SDS3 injection into MMTV-PyMT; ACTB-ECFP mice shows SDS3 present within tumor stroma (n = 8). See Video 1 for full video of still shots shown; bottom—metastatic lung confocal microscopy 24 h after SDS3 injection into MMTV-PyMT; ACTB-ECFP mice shows SDS3 accumulating at the metastatic sites as compared with lung parenchyma (n = 5). See Video 2 for full video of still shots shown. **(B)** An MMTV-PyMT; ACTB-ECFP mouse was i.v. injected with VO-PyMT-GFP-Luc cells. 1 wk later, SDS3-HyLite 555 was injected 24 h before imaging followed by near-infrared (NIR) 10-kD dextran and anti-Gr1 antibody 5 h before imaging. Representative images of confocal microscopy show NIR 10-kD dextran and anti-Gr1 antibody (white) accumulate around VO-PyMT metastasis (green) and SDS3-HyLite 555 (red) (n = 5). See Videos 3 and 4 for full video of still shots shown. **(C)** A representative flow cytometry analysis of the lungs of MMTV-PyMT mice 24 h after i.v. injection of VO-PyMT-GFP-Luc cells and treated with SDS3 or IgG isotype control. **(D)** Left: 1 × 10^5 VO-PyMT-GFP-Luc cells were i.v. injected into MMTV-PyMT mice along with SDS3-HyLite 750. MMTV-PyMT mice IVIS image depicts high intensity of signal resting within the lungs 24 h after injection (n = 3). Right: 1 × 10^6 VO-PyMT-GFP-Luc cells were i.v. injected into WT mice along with SDS3-HyLite 750. Fluorescent IVIS whole-body imaging shows localization of SDS3-HyLite 750 and bioluminescent IVIS whole-body imaging shows VO-PyMT-GFP-Luc cells seeding within the lungs of WT mice (n = 3). **(E)** Ex vivo fluorescent and bioluminescent imaging of various organs from WT mice i.v. injected with 1 × 10^6 VO-PyMT-GFP-Luc shows that the strongest signal was detected in the lung, indicating SDS3 accumulation at the metastatic site with no retention of IgG isotype control within the lungs (n = 3 IgG-HyLite 750, n = 3 SDS3-HyLite 750).

gelatinolytic activity compared with control mice (Figs 4C and D, and S4B).

We next determined if blocking MMP9 had direct effects on the tumor cells *in vitro*. Although SDS3 treatment had no effect on VO-PyMT cell viability *in vitro* (Fig 5A), in a two-dimensional scratch assay, SDS3 significantly inhibited migration (Figs 5B and S5A). Although *Mmp2* was elevated, this did not compensate functionally for the inhibition of MMP9. *Mmp3*, a potent activator of MMP9

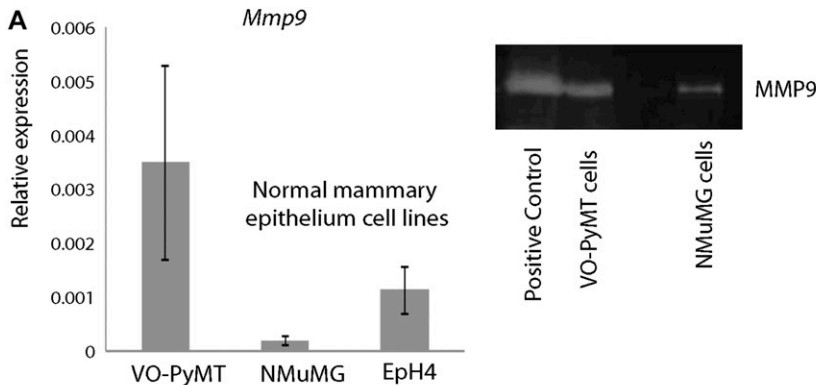

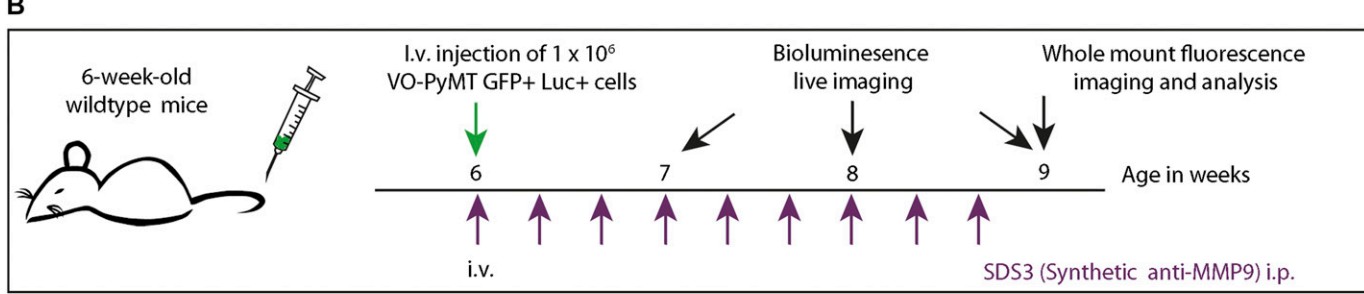

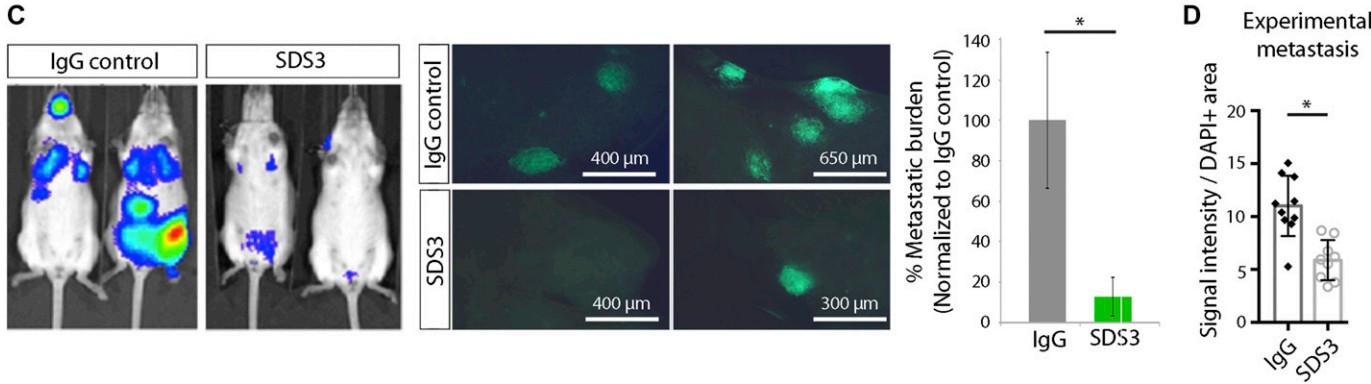

**Figure 4. SDS3 inhibits experimental lung metastases from circulating, MMP9-producing VO-PyMT metastatic cells.**
**(A)** Left: qPCR of *Mmp9* levels in VO-PyMT-GFP-Luc cells and the normal mammary cell lines EpH4 and NMuMG. Right: gel zymography of indicated cell lines demonstrating MMP9 activity. **(B)** Schematic of experimental setup: 1 × 10⁶ VO-PyMT-GFP-Luc cells along with IgG isotype control or SDS3 was i.v. injected into WT mice. Subsequently, the antibodies were i.p. injected every other day for eight times before the mice were euthanized. Weekly bioluminescence imaging was performed to monitor the growth of the i.v.–injected tumor cells. **(C)** Left: bioluminescent imaging of 9-wk-old WT mice 3 wk after i.v. injection of VO-PyMT-GFP-Luc probing cells. Middle: representative fluorescent whole lung images of WT mice treated with SDS3 or IgG after a 3-wk chase. Right: quantification of the lung metastatic burden in WT mice treated with SDS3 or IgG (n = 9 IgG, n = 9 SDS3). **(D)** Quantification of in situ zymography of lungs in WT mice treated with SDS3 or IgG (n = 3 each).

(Vempati et al, 2007), was elevated but was not sufficient to rescue the phenotype. Tenascin C (*Tnc*), a gene expressed in cancer cells on the invasive front (Lowy & Oskarsson, 2015), was down-regulated post-SDS3 with no change in *caspase 3*, a pro-apoptotic marker. In addition, there were no noticeable changes in tumor suppression, proliferation, or epithelial-to-mesenchymal transition (EMT), as reflected by *Loxl4*, *periostin*, and *E-cadherin* expression levels (Fig 5C). We confirmed that SDS3 inhibited ECM invasion using a Matrigel-coated trans-well invasion assay (Figs 5D and S5B). Although VO-PyMT cell viability remained unaffected (Fig 5A), we observed a down-regulation of proliferation markers, including *c-Myc* and *Ki67* (Fig 5E).

To validate our *in vitro* characterization of migration and invasion, we used *ex vivo* confocal microscopy on the lungs of MMTV-PyMT mice 24 h after injecting SDS3-HyLite 555 or IgG control. Time-lapse imaging illustrated reduced migration of VO-PyMT cells after injection of SDS3 (Fig 5F, Videos 5 and 6). These data further confirm that SDS3 attenuates the ability of VO-PyMT cells to migrate and subsequently invade into the lungs.

Finally, we examined whether SDS3 affected the stem-like, colony formation characteristics of VO-PyMT cells, which are critical for metastatic colonization. SDS3 treatment significantly reduced the number and size of colony-forming spheres when

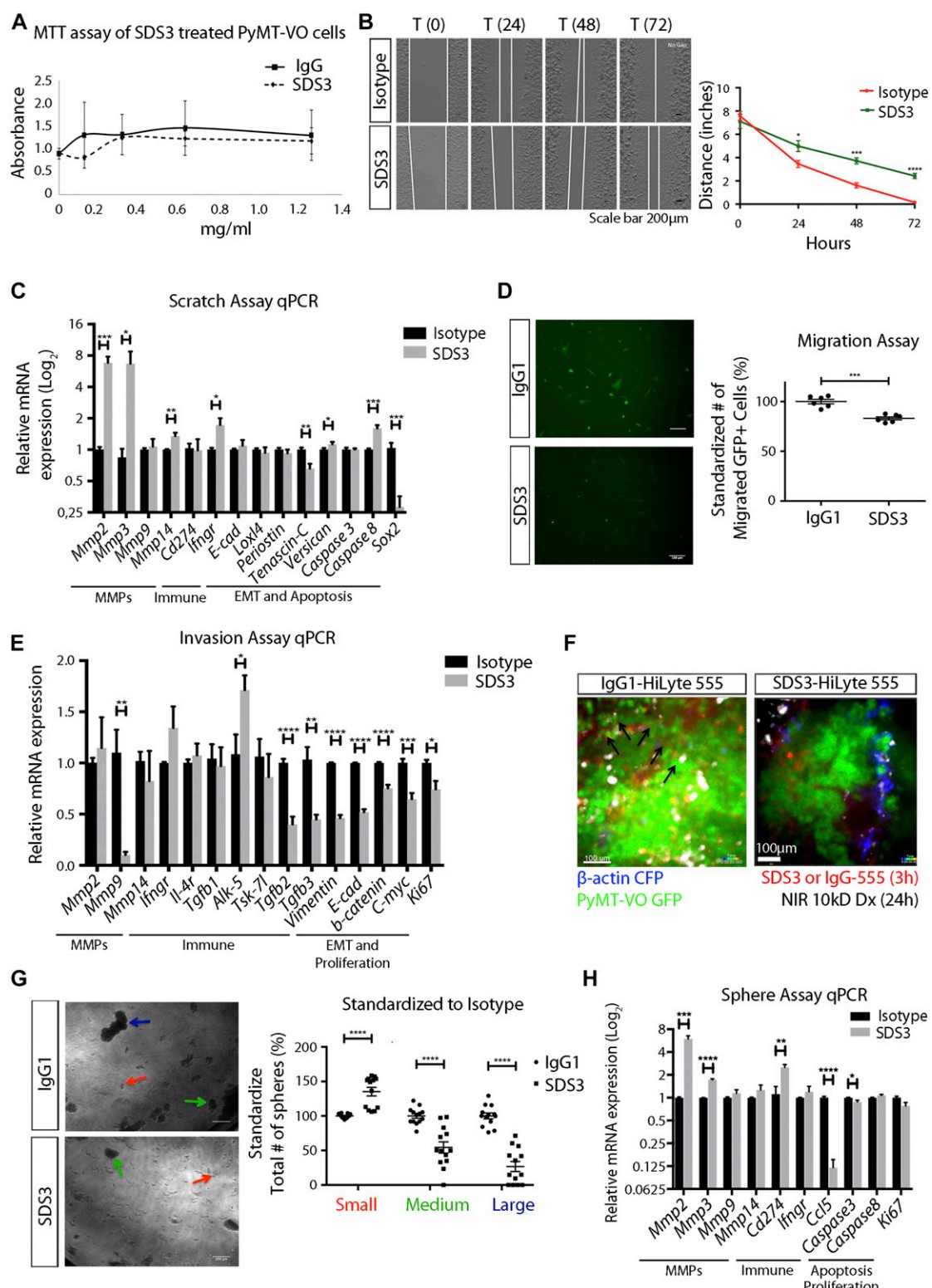

**Figure 5. SDS3 blocks migration, tissue invasion, and colony formation of MMP9-producing VO-PyMT tumor cells.**
**(A)** MTT assay of VO-PyMT cells treated with 0, 0.125, 0.3125, 0.625, or 1.25 mg/ml of SDS3 or IgG isotype control. The selected concentrations are equivalent to treating mice with 0, 1, 2.5, 5, and 10 mg/kg of antibody, respectively. **(B)** Left: representative images of scratch assay. VO-PyMT cells were treated with 0.625 mg/ml of IgG or SDS3. Area in between the white lines depict the width of area that cells have not migrated back into. Right: quantification of the scratch edge distance (between white lines) after treatment with IgG or SDS3 at the indicated time points (n = 4 IgG, n = 4 SDS3). **(C)** qPCR analysis of indicated targets in VO-PyMT cells after scratch and 72 h of IgG or SDS3 treatment. Triplicate conditions were performed for each target with two biological replicates. **(D)** Left: trans-well migration assay of VO-PyMT, GFP⁺ cells in vitro

compared with IgG control (Figs 5G and S5C). In line with the expression profile seen with the scratch assay, we found that *Mmp2* and *Mmp3* were up-regulated, whereas *Mmp9* levels were unaffected (Fig 5H). Taken together, our data demonstrate that MMP9 inhibition with SDS3 attenuated migration, invasion, and colony-forming ability independent of regulating *Mmp9* mRNA levels.

### Functional inhibition of MMP9 enhances CD8⁺ T cell invasion and activation at sites of metastasis

Our analysis of invasiveness also suggested potential modulation of the immune landscape after inhibition with SDS3. Specifically, we observed (Fig 5E) (i) decreased *Tgfβ2* and *Tgfβ3*, which regulate invasiveness, (ii) a trend towards an anti-tumorigenic cytokine profile with up-regulation of *Ifngr* and *Cd274* (which encodes programmed death-ligand 1 [PD-L1]), and (iii) an increase in *caspase 8*, which activates a signaling cascade to recruit activated T cells (Fig 5C and H) (Newton & Strasser, 2003; Plaks et al, 2015). Previous work has demonstrated that activation of the immune system causes subsequent up-regulation of PD-1/PD-L1 and is thought to be partially induced by IFNγ (Keir et al, 2008). Based on our findings, we hypothesized that SDS3 treatment may affect the metastatic microenvironment by tilting the balance towards Th1 immunity.

We then examined changes in immune function before changes in metastatic burden, 1 wk after SDS3 treatment. Interestingly, we observed significant activation of T cells in both the primary tumor and lungs, characterized by elevated inducible T-cell costimulator (ICOS⁺) and CD25⁺ cells, with no differences in the number of regulatory T cells (Tregs) (Figs 6A and S6A–D). Although we saw an activation of immune cells present within the primary tumor, histological sections of the mammary glands showed no noticeable difference in tumor stage and progression (Fig S6E). Although both CD4⁺ and CD8⁺ T cells had elevated levels of ICOS (Figs 6B and S7A), only CD8⁺ICOS⁺ T cells exhibited elevated levels of IFNγ, a key cytokine mediating the cytotoxic activity of CD8⁺ T cells (Figs 6C and D and S7B). These activated CD8⁺ T cells also exhibited significantly elevated PD-1, which indicate recent T-cell activation (Figs 6E and F and S7C and D) (Keir et al, 2008). Furthermore, we observed changes in the myeloid population (Fig S7E and F), including a significant increase in neutrophils, which is associated with T-cell activation (Uribe-Querol & Rosales, 2015).

Since MMP9 is an activator of TGFβ, which directly impacts T-cell responses (Juric et al, 2018; Tauriello et al, 2018), we investigated whether SDS3 treatment affects the effector T-cell compartment. Our *in vitro* analyses (Fig 5) on tumor cells showed an up-regulation of *Cd274*, associated with activation of the immune system (Keir et al, 2008), and down-regulation in *Ccl5*. Interestingly, qPCR of MMTV-PyMT lungs 1-wk post-treatment showed reduction in *Tgfβ1* and *Alk-5*, a subunit of *Tgfβ1r*, with an elevation in *interleukin-12* (*IL-12*), a proinflammatory Th1 cytokine, suggesting that SDS3 abrogated immunosuppression (Fig 6G). These results suggest that MMP9 regulates aspects of the immune response, specifically, elevation of IFNγ, inducing a shift towards an anti-tumorigenic lung microenvironment. Treatment with SDS3 enhanced CD8⁺ T-cell infiltration into the metastatic lung, concentrating around metastatic foci (Figs 6H and S7G and H). Overall, these data demonstrate that blocking MMP9 using SDS3 affected not only multiple intrinsic properties of tumor cells but also affected recruitment and activation of CD8⁺ T cells in the lung microenvironment, which ultimately contributed to an anti-metastatic immune response (Fig 7).

## Discussion

In this study, we demonstrate that a metastatic niche is established early during tumorigenesis in MMTV-PyMT, an autochthonous, immune competent mouse model of luminal B BC. This metastatic niche was driven, at least in part, by MMP9, which was supplied by CD11b⁺Gr1⁺ myeloid cells, among other cells, which establish the lung metastatic niche (Kowanetz et al, 2010; Casbon et al, 2015; Wculek & Malanchi, 2015). This metastatic niche facilitates the colonization and growth of CTCs. Interestingly, once clusters of tumor cells are formed in the lungs, these myeloid cells accumulate predominantly around the metastatic foci.

We show that SDS3, a unique antibody that targets the active site of gelatinases, had anti-metastatic activity and primarily suppressed metastatic growth in the lungs, without significantly affecting primary tumor burden. Specifically, SDS3 mediated inhibition of tumor cell–intrinsic properties, such as migration, invasion, and colony-forming ability, which are required for metastatic colonization. In addition, SDS3 led to an enhanced immune response by increasing recruitment and activation of CD8⁺ cytotoxic T cells around metastatic foci. In examining the effect of SDS3 on the primary tumor, once tumor foci had been established within the TME, MMP9 no longer played a critical role in tumor growth, rendering SDS3 ineffective (Fig 7). Our results showed that MMP9 is necessary for the priming of the metastatic niche to enable metastatic growth, but further work is required to uncover when metastatic cells in our model become independent of MMP9 supplied by the stroma *in vivo*.

Our study suggests that MMP9 inhibition may be an important strategy in controlling the establishment of the metastatic niche, predominately early during tumorigenesis before widespread dissemination of tumor cells into circulation. This supports the

---

treated with 0.375 mg/ml of SDS3 or IgG for 24 h. Right: quantification of the number of cells migrating through a Matrigel-lined insert after SDS3 or IgG treatment/incubation. (n = 6 IgG, n = 6 SDS3). **(E)** qPCR analysis on IgG or SDS3-treated trans-well invasion assay VO-PyMT, GFP⁺ cells 24 h posttreatment. Significant down-regulation of *Mmp9* captured using a 3D invasion assay, whereas no change in *Mmp2* provides evidence of SDS3 specifically targeting MMP9. Triplicate conditions performed for each gene with two biological replicates. **(F)** Ex vivo confocal microscopy of MMTV-PyMT mice showing migrating VO-PyMT, GFP⁺ cells (arrows) in mice 24 h after injection with IgG-HyLite555 as compared with none in mice injected with SDS3-HyLite 555. See Videos 5 and 6 for full video of still shots shown. **(G)** Left: representative images of a colony formation assay using VO-PyMT cells (4,000 cells/well) treated with 0.625 mg/ml of IgG or SDS3. Arrows depict the categorization of small (red), medium (green), and large (blue) spheres. Right: quantification of sphere size in each well (n = 2 biological replicates with seven replicates per condition and five images per well quantified). **(H)** qPCR analysis on IgG- and SDS3-treated sphere assay of VO-PyMT cells 5 d posttreatment. Elevation of *Mmp2* but no noticeable change in *Mmp9* transcript levels supports the specificity of SDS3. Triplicate conditions performed with two biological replicates.

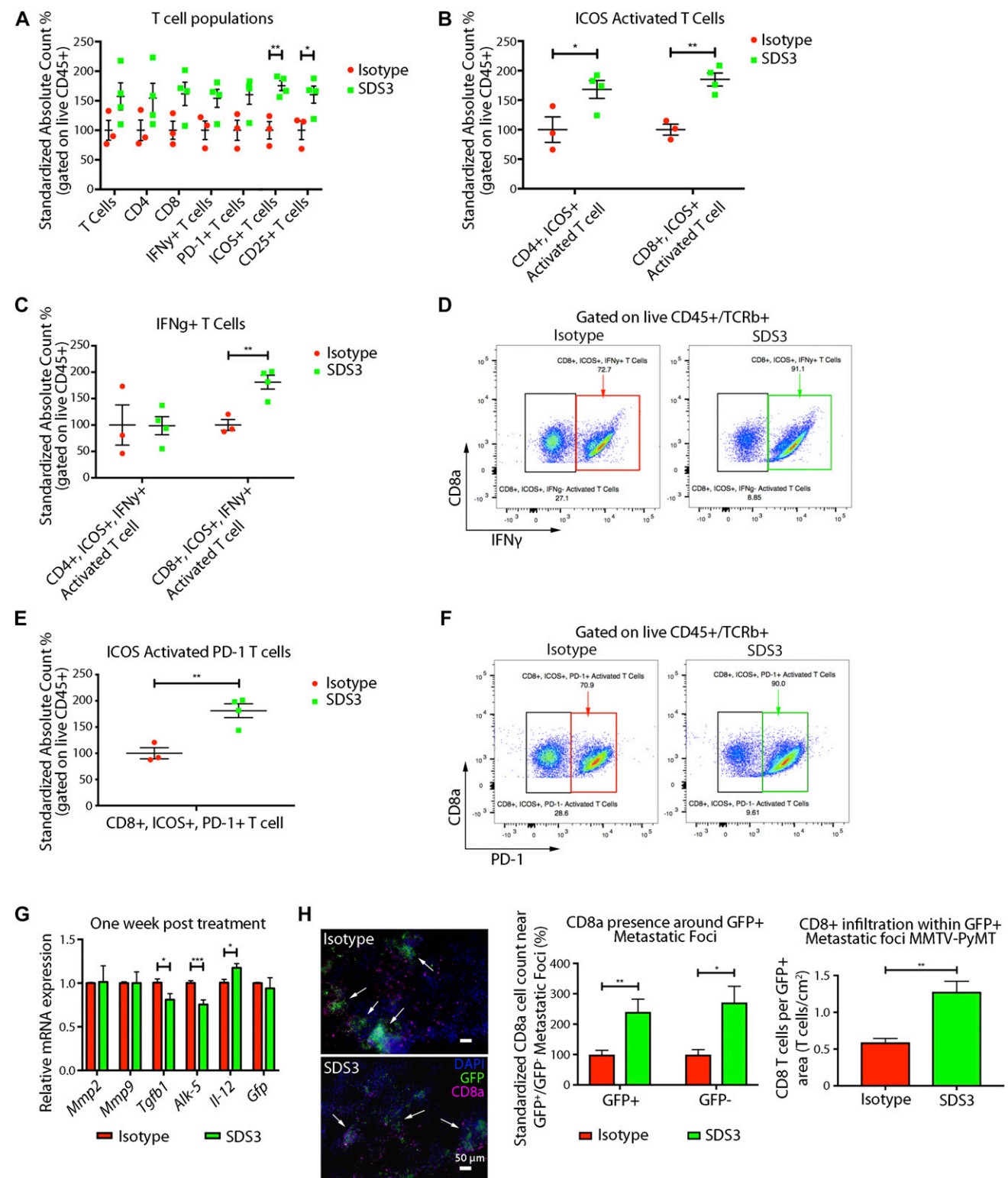

**Figure 6. SDS3 promotes CD8⁺ T cell activation and T cell infiltration.**
**(A)** Flow cytometry analysis of early immune changes within the lungs of MMTV-PyMT mice i.v. injected with 1 × 10⁵ VO-PyMT-GFP-Luc cells at 7 wk of age treated with SDS3 or IgG isotype control. Counting beads used to normalize frequencies with absolute cell counts standardized to isotype control expressed as a relative difference from the average isotype control absolute cell count. Significant activation of T cells seen through marked elevation of ICOS and CD25 (n = 3 IgG, n = 4 SDS3). **(B)** Flow cytometry analysis of ICOS-activated T cells reveals both CD4 and CD8 T cell subsets significantly up-regulated after SDS3 treatment. **(C)** Flow cytometry analysis of IFNγ within CD4- and CD8-activated T cells (ICOS+) demonstrates prominent Th1 cytokine skewing of CD8 T cells after SDS3 treatment. **(D)** Representative scatterplot of ICOS+

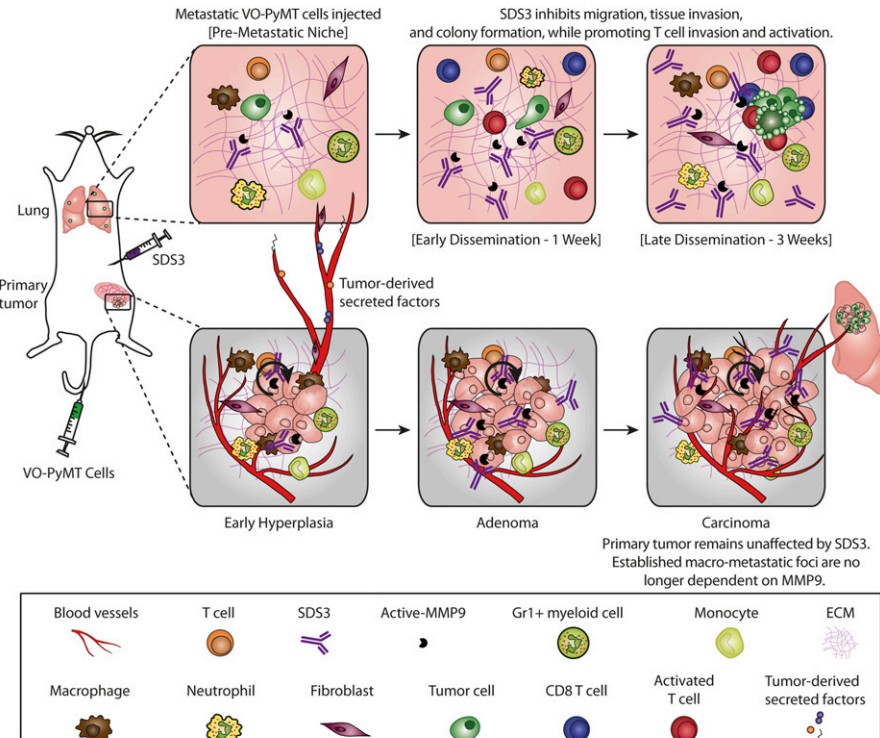

**Figure 7. Summary of SDS3's effect within the TME and changes in immune architecture after inhibition of active MMP9.**

notion that tumor cell dissemination occurs early during BC progression (Lambert et al, 2017). Therefore, identifying these early niche changes in human patients may yield additional therapeutic strategies to block metastatic cell colonization and proliferation at distant sites. One of the major drawbacks of studies thus far that have investigated early metastatic niche formation is utilization of late-stage and extremely aggressive tumor cell line transplants (e.g., B16 melanoma, Lewis lung carcinoma, and 4T1 breast carcinoma) into mice. These models do not capture the initial changes that occur early during tumor development, and thus, it remains unclear whether the factors identified thus far play roles in early tumorigenesis. From a clinical perspective, this distinction is critical because early tumors (or recurring hyperplasia) may secrete factors that predispose the patient to metastatic colonization years later. This suggests that some patients may benefit from the therapy to control or delay the onset of occult metastases, even when there is no radiographic evidence to suggest disseminated or metastatic disease. With some clinical trials now using metastasis-free survival as a clinically meaningful endpoint (Smith et al, 2018), the ability to inhibit and delay metastasis is important.

In early stages of metastatic colonization, we postulate that there is a minimum threshold of active MMP9 within the TME required for the priming of the premetastatic niche to be more susceptible for colonization. We also showed that the primary tumor could also provide MMP9 that could be responsible for invasion and metastasis. Within the MMTV-PyMT model, neutrophils have been recently implicated as principal players in cancer initiation and progression (Coffelt et al, 2016). More specifically, Alox5 expression can promote downstream metastatic effects through induction of MMP9 (Kummer et al, 2012). We observed MMP9-mediated priming of the metastatic niche within the MMTV-PyMT mouse model and showed that MMP9 (and not MMP2) dominates metastasis in human BC and in the MMTV-PyMT model with genetic ablation of MMP9. Interestingly, exosomal MMPs have been thought to also assist in the proteolytic remodeling process of the ECM through activation of MMP2 (Shay et al, 2015).

In addition, T cells can be inactivated by inflammatory cells in the microenvironment (Grivennikov et al, 2010). Identifying and targeting suppressive signals could be impactful in remodeling the immune landscape to tilt the balance towards an anti-tumorigenic

CD8 T-cell expression of IFNγ. Red box represents IgG-treated activated CD8 T cells expressing IFNγ. Green box represents SDS3-treated activated CD8 T cells expressing IFNγ. **(E)** Flow cytometry analysis of PD-1 within CD8[+]-activated T cells (ICOS+) shows significant up-regulation of an exhausted T-cell response due to overactivation of the immune system. **(F)** Representative scatterplot of ICOS+ CD8 T-cell expression of PD-1. Red box represents IgG-treated activated CD8 T cells expressing PD-1. Green box represents SDS3-treated activated CD8 T cells expressing PD-1. **(G)** qPCR analysis of MMTV-PyMT lungs i.v. injected with 1 × 10[5] VO-PyMT-GFP-Luc cells 1 wk after IgG or SDS3 treatment shows decreased Th2/M2-like markers (*Il-4r* and *Egr2*) and increased *Il-12* (duplicate conditions with n = 4 IgG and n = 4 SDS3). **(H)** Left: representative immunofluorescence images of 9-wk-old MMTV-PyMT lungs i.v. injected with 1 × 10[5] VO-PyMT-GFP-Luc cells 3 wk after IgG or SDS3 treatment. CD8[+] T cells localize around GFP[+] metastatic sites (arrows) after IgG and SDS3 treatment, whereas lower presence of CD8[+] T cells seen around GFP[−] sites. Middle: quantification of CD8[+] T cells in GFP[+] and GFP[−] regions 3 wk after IgG and SDS3 treatment (not shown: GFP[−] foci). Right: CD8[+] T cell infiltration around periphery and within GFP[+] metastatic foci in MMTV-PyMT mice at 9-wk of age after a 3-wk chase post i.v. injection of 1 × 10[5] VO-PyMT GFP[+] cells (n = 5 IgG, n = 5 SDS3; three serial sections used for quantification).

TME, ultimately prolonging overall survival. Interestingly, TGFβ is associated with resistance to anti–PD-L1 by promoting T-cell exclusion in bladder cancer (Lau et al, 2017) and colorectal cancer (Tauriello et al, 2018). Targeting TGFβ alone showed efficacy against a metastatic disease but had no effect on the primary tumor (Lau et al, 2017; Tauriello et al, 2018). Because MMP9 is a known activator of TGFβ (Tauriello et al, 2018), therapy targeting active MMP9 in combination with checkpoint inhibitors is warranted as a future direction. With recent developments in bifunctional approaches to perturb the immune system, specifically using TGFβ/PD-L1 (David et al, 2017; Lan et al, 2018; Ravi et al, 2018), modulating the ECM in cancers currently resistant to checkpoint inhibitor monotherapy may increase the efficacy.

Understanding mechanisms to modulate the microenvironment for the optimization of checkpoint blockade therapy are currently underway (Mariathasan et al, 2018; Owyong et al, 2018). For example, the effects of an anti-MMP9 monoclonal antibody (GS-5745) are being examined in combination with nivolumab (anti-PD-1) and other anticancer drugs for recurrent gastric and gastroesophageal adenocarcinoma (clinicaltrials.gov). This suggests the possibility to modulate the microenvironment during early metastatic growth to get better responses in BC patients who generally arrive at the clinic with an established primary tumor and have developed early metastases to form micrometastatic foci. In their totality, our data demonstrate that regulation of the ECM by targeting MMP9 alters the hallmark metastatic processes of migration, invasion, and colony formation and remodels the immune landscape, redefining MMP9 as an immune modulator for BC therapy. Taken together, targeting MMP9 may be an effective combination therapy and suggests that perturbing the MMP9–T-cell axis may provide a new strategy to overcome resistance to immunotherapy.

# Materials and Methods

### Animal studies

All animal experiments were performed at University of California, San Francisco (UCSF) and reviewed and approved by UCSF Institutional Animal Care and Use Program. Mice were housed under pathogen-free conditions in the UCSF barrier facility. FVB/n mice, originally from Charles River, were bred in-house or purchased through Charles River. For premetastatic niche setup, 6-wk-old MMTV-PyMT female mice were i.v. injected with $1 \times 10^5$ VO-PyMT probing cells in PBS. For experimental metastasis experiments, 6-wk-old WT female mice were i.v. injected with $1 \times 10^6$ VO-PyMT probing cells in PBS. For genetic examination of total MMP9, MMTV-PyMT; MMP9 mice were crossed. Therapeutic antibody was i.p. injected using 5 μg/g (per injection) using SDS3, HyLite 750 conjugated SDS3 (IVIS imaging), HyLite 555 conjugated SDS3 (confocal imaging), or a nonspecific isotype (IgG1, MOPC-21). Dr. Irit Sagi's group prepared SDS3 as described previously (Sela-Passwell et al, 2011).

Tumor measurements were made using a caliper once per week. Bioluminescent imaging was performed using an IVIS Spectrum and image radiance normalized using Living Image (Caliper LifeScience). For whole-mount fluorescent imaging, mouse lung lobes were squeezed between two glass slides, removing the volumetric dimension to achieve area in square millimeter using ImageJ.

### Cell culture

The PyMT cell line VO-PyMT (Halpern et al, 2006) was a gift from Conor Lynch (Moffitt Cancer Center). The LAP0297 cell line, generated from a spontaneous lung carcinoma in FVB/n mice (Huang et al, 2008), was a gift from Peigen Huang (Massachusetts General Hospital). The cells were transduced with pMSCV-luciferase to generate a luciferase-expressing cell line and with pMIG to generate a GFP-expressing cell line. Cell sorting was performed on a FACS Aria II (Becton Dickinson), and the cells were grown in DMEM, high glucose (11965; Gibco) supplemented with 10% FBS and 1% penicillin–streptomycin.

### Retroviral production

Viral production was carried out using calcium phosphate–mediated transfection of HEK293T/GP2 cells. Virus was concentrated by ultracentrifugation and added to cells with polybrene. Stably transduced cells were selected in puromycin for at least 5 d or selected by FACS.

### qPCR

Total RNA was isolated from cells using the RNeasy Mini Kit (QIAGEN). cDNA was synthesized using the Superscript III RT First Strand Kit (Invitrogen). qPCR was performed using FastStart Universal SYBR Green master mix (Roche) in an Eppendorf Mastercycler Realplex machine. Ct values were normalized to GAPDH and actin, and relative expression was calculated using the $2^{-DDCt}$ method. Primer sequences for qPCR were found using the Harvard Primer Bank and are detailed in Table S1.

### Immunostaining and histology

Tissues were fixed in 4% PFA overnight, paraffin processed or embedded into OCT for frozen sections, and sectioned. Hematoxylin and eosin staining was performed for routine histology. Antigen retrieval was performed using citrate or proteinase K for immunohistochemistry. The Tyramide Signal Amplification Kit (#NEL700A001KT; Perkin Elmer) was used according to the manufacturer instructions. Primary antibodies were incubated overnight and secondary antibodies were incubated for 1 h. The following antibodies were used: phosphohistone H3 (#9701, 1:100; Cell Signaling), CD8a (16-0081, 1:500; eBioscience), GFP (ab290, 1:500; Abcam), biotinylated antirat (#112-067-003, 1:300; Jackson), biotinylated antirabbit (#E0431, 1:300; Dako), and biotinylated antigoat (#305-067-003, 1:300; Jackson). Fluorescent antibodies 488-antirabbit, 568-antirat, and 647-antirat were from Molecular Probes (#A11008, #A11077 and A-21247, 1:600; Invitrogen), with fluorescent microscopy performed on a Keyence BZ-X700. Image analysis was performed using ImageJ.

### In situ zymography and immunofluorescence

Gelatinolytic activity was assessed by in situ zymography of zinc-buffer fixative (36.7 mM $ZnCl_2$, 27.3 mM $ZnAc_2 \times 2H_2O$, and 0.63 mM

CaAc$_2$ in 0.1 mol/l Tris, pH 7.4), paraffin-embedded sections of lungs and mammary gland from 6- to 10-wk-old MMTV-PyMT mice. Sections were heated at 59°C for 3 h, deparaffinized in xylene, and rehydrated in graded alcohol baths. Substrate was prepared by dissolving 1 mg DQ gelatin (Molecular Probes; Thermo Fisher Scientific) in 1.0 ml Milli-Q water and further diluted 1:50 in a reaction buffer containing 50 mM Tris–HCl, 150 mM NaCl, and 5 mM CaCl$_2$ (pH 7.6). Sections were covered with the substrate solution and parafilm, and then incubated for 2 h in a dark, humid chamber at 37°C. The parafilm was removed, then the sections were rinsed with Milli-Q water, and washed in PBS (2 × 5 min). The sections were then incubated with DAPI (diluted 1:10,000; Molecular Probes/Thermo Fisher Scientific) for 5 min, washed in PBS baths (2 × 5 min), and mounted with Fluorescent Mounting Medium (Dako). Contribution of MMPs to the observed gelatinolytic activity and the level of auto fluorescence in the tissues was assessed by incubating parallel sections with 20 mM EDTA added to the substrate solution or without DQ gelatin added to the reaction buffer, respectively. In some sections, the SDS3 antibody was added to the substrate solution (diluted 1:50) to assess contribution of MMP9 to the gelatinolytic activity. Fluorescent microscopy performed on Keyence BZ-X700 with 10× magnification at randomly selected fields and quantified using ImageJ by measuring the inner and outer ring integrated density of the GFP area standardized to WT of each age group.

7-wk time point WT and MMTV-PyMT sections were immuno-fluorescently stained for MMP2, MMP9, CD3, F4/80, and Gr1 after in situ zymography procedures (before DAPI staining) to determine if gelatinolytic activity co-localized with these enzymes. After substrate incubation and rinsing, the sections were incubated for 1 h at room temperature with a blocking solution containing 5% goat serum, 0.1% Triton X-100, and 0.05% Tween 20 in PBS to block nonspecific binding of the antibody. Sections were then incubated for 1 h (room temperature in a dark, humid chamber) with primary antibody (MMP2 [ab37150; Abcam, diluted 1:50 in blocking solution] or MMP9 [ab388898; Abcam, diluted 1:500 in blocking solution]), rinsed in PBS with 0.05% Tween20 (PBST) 5 min × 3, and incubated for 1 h (room temp. in a dark, humid chamber) with secondary antibody (Alexa Fluor 633 [AF633], goat antirabbit IgG [Molecular Probes, diluted 1:400 in blocking solution]). These sections were also incubated with primary antibody (F4/80 [Molecular Probes MF4800 diluted 1:20 in blocking buffer], CD3 [MAC2690T; Bio-Rad diluted 1:40 in blocking buffer], and Gr1 [MA1-70099; Thermo Fisher Scientific diluted 1:20 in blocking buffer]) at 4°C overnight. They were rinsed in PBST for 3 × 5 min and subsequently incubated with secondary antibody (AF 633, goat antirat IgG [A21094; Invitrogen, diluted 1:200 in blocking buffer], DyLight649, goat anti-Armenian hamster IgG [Jackson Immuno Research code 127-495-160, diluted 1:200 in blocking buffer]) for 1 h at room temperature followed with rinsing in PBST 3 × 5 min. The sections were then incubated with DAPI (Molecular Probes/Thermo Fisher Scientific) (diluted 1:10,000) for 5 min, washed in PBS baths (2 × 5 min), and mounted with Fluorescent Mounting Medium (Dako). Fluorescent microscopy was performed on Keyence BZ-X700. For CD8a quantification, a circle of defined radius from the perimeter of the GFP$^+$ region was defined before counting all CD8a$^+$ cells that sit within the defined region of interest.

## In vitro assays

### Sphere assay

$1 \times 10^4$ VO-PyMT cells were suspended within complete Matrigel (CB40234; Corning) and seeded into a round bottom 96-well plate. DMEM, high glucose (11965; Gibco) with 10% FBS, and 1% penicillin–streptomycin were added to each well and changed daily. 0.375 mg/ml of SDS3 and IgG control used per well. Images were taken using inverted microscope and Leica software at day 0, 4, 5, and 6. Spheres were graded based on size and categorized into small, medium, large, and branching percentages per well.

### Migration assay

VO-PyMT, GFP$^+$ cells were suspended at $1 \times 10^4$ cells per 200 $\mu$l of DMEM, high glucose (11965; Gibco) with 1% penicillin–streptomycin and 0.5% puromycin. 700 $\mu$l of DMEM with 2% FBS, 1% penicillin–streptomycin, and 0.5% puromycin along with 0.375 mg/ml of IgG control or SDS3 was added into each well of a 24-well flat bottom plate. FluoroBlok inserts (351152; Corning) were lined with 25 $\mu$l of 1:2 diluted growth factor reduced Matrigel (354230; Corning) and incubated for 45 mins at 37°C. VO-PyMT, GFP$^+$ cells were seeded into FluoroBlok insert and imaged 24 h postseeding. Images were taken with an inverted fluorescent microscope for GFP$^+$ signal using Leica software. Each well was split into four quadrants and a random image was taken within each quadrant at 10× magnification. The number of migrated, GFP$^+$ cells were counted and compared against the total number of cells seeded.

### Scratch assay

VO-PyMT cells plated onto six-well plates at $5 \times 10^5$ cells/2.5 ml and allowed for form a confluent monolayer for 48 h before T(0). 0.625 mg/ml of IgG or SDS3 was added into each well at T(0) with DMEM, high glucose, 10% FBS, and 1% penicillin–streptomycin. Images were taken on a Keyence microscope every 24 h. The distance between the white lines represents the area that VO-PyMT cells have not migrated to. Quantification conducted by measuring the distance between the white lines.

## Flow cytometry

To process lungs for FACS, we used 50 $\mu$l Liberase (26 units/ml, 5401127001; Sigma-Aldrich) and 50 $\mu$l DNase (10 mg/ml, D4263; Sigma-Aldrich) per animal suspended in 5 ml DMEM, high glucose. The samples were digested using gentleMACS C tubes (130-093-237; MACS), then incubated after initial homogenization on an oribital shaker for 30 min at 37°C, 150 rpm (~0.2$g$). Lung samples were lysed with RBC lysis buffer and then filtered through a 70-$\mu$m cell strainer before staining. Fluorescent antibodies used for FACS are listed in Table S2.

For cytokine stimulation, single cell suspension was stimulated with PMA+ionomycin without Brefeldin A (423301; BioLegend) for 4 h according to the manufacturer's instructions and BD GolgiPlug added 1 h after initial stimulation. After extracellular staining, the cells were fixed and permeabilized using the BD Cytofix/Cytoperm Plus (555028; BD Biosciences) and then stained for intracellular markers.

## Confocal microscopy

Details of the microscope design as well as a detailed protocol for ex vivo lung imaging were previously described (van den Bijgaart et al, 2016). To image lung capillaries, mice were i.v. injected with 100 μl sterile PBS containing 4 mg/ml 10 kD AF647-conjugated dextran or 4 mg/ml 70 kD rhodamine-conjugated dextran (Invitrogen). In addition, to image Gr1+ neutrophils and monocytes, mice were i.v. injected with 100 μl sterile PBS containing 1 mg/ml AF647-conjugated Gr-1 antibody 5 h before excision of the lungs. At the time of lungs' excision, the mice were euthanized by i.p. injection of 1 ml of 2.5% Avertin. Subsequently, the lungs were inflated with 400 μl of 37°C 2% low melting temperature agarose followed by excision of the lungs. The lungs were immersed in RPMI-1640 and the lobes were gently separated and then transferred to an imaging plate.

## Bioinformatics analysis of illumina microarray

Public microarray datasets were downloaded from the National Center for Biotechnology Information, Gene Expression Omnibus. The MMP family mRNA expression levels were analyzed across three normal samples and 106 metastasis samples as well as six positive and five negative controls in GSE32490 dataset. Probe intensities were analyzed and normalized using the Lumi package in the R statistical environment as previously described (Du et al, 2008).

## Statistical analysis

Statistical analysis was performed using Prism 7 software (GraphPad Software, Inc.). All data are presented as mean ± SEM, unless otherwise stated. When two groups were compared, the two-tailed $t$ test was used, unless otherwise stated. When three or more groups were compared, the one-way ANOVA test was used, followed by Tukey's test to determine significance between groups. We considered $P < 0.05$ as significant.

# Supplementary Information

# Acknowledgements

We thank members of the Werb laboratory for helpful discussions. We specifically thank Elena Atamaniuc, Ying Yu, Kiarash Salari, Ankitha Nanjaraj, and Helen Capili for technical assistance; Caroline Bonnans for immuno-fluorescence advice; Nguyen H Nguyen and Vaishnavi Sitarama for their assistance with therapeutic experiments; and Catharina Hagerling and the University of California, San Francisco Flow Cytometry Core for advice with flow cytometry. This work was supported by a Department of Defense Postdoctoral Fellowship (W81XWH-11-01-0139) to V Plaks, Department of Defense Predoctoral Fellowship (W81XWH-10-1-0168) to J Chou, grants from the National Cancer Institute (R01 CA057621, U01 CA199315, CA180039, and CA190851) and the Parker Institute for Cancer Immunotherapy to Z Werb, grants from the California Breast Cancer Research Program (23IB-001) and Cancer League Award to Z Werb and V Plaks, and funds from the Israel Science Foundation (1800/19), the USA-Israel Binational Science Foundation (712506-01), the European Union's Horizon 2020 research and innovation program (grants agreement No [801126] and [695437]), and The Thompson Family Foundation, Inc. to I Sagi.

## Author Contributions

M Owyong: conceptualization, data curation, formal analysis, validation, investigation, visualization, methodology, and writing—original draft, review, and editing.
J Chou: conceptualization, data curation, investigation, methodology, and writing—original draft, review, and editing.
RJE van den Bijgaart: data curation, investigation, visualization, methodology, and writing—original draft, review, and editing.
N Kong: formal analysis.
G Efe: formal analysis, investigation, and methodology.
C Maynard: formal analysis.
D Talmi-Frank: methodology.
I Solomonov: methodology.
C Koopman: formal analysis.
E Hadler-Olsen: formal analysis and methodology.
M Headley: resources and data curation.
C Lin: resources.
C-Y Wang: software and formal analysis.
I Sagi: supervision, funding acquisition, and writing—review and editing.
Z Werb: conceptualization, supervision, funding acquisition, investigation, and writing—original draft, review, and editing.
V Plaks: conceptualization, data curation, funding acquisition, investigation, and writing—original draft, review, and editing.

## Conflict of Interest Statement

The authors declare that they have no conflict of interest.

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
