## [Reviewer comments · Life Science Alliance]

Life Science Alliance

MMP9 Modulates the Metastatic Cascade and Immune Landscape for Breast Cancer Anti-Metastatic Therapy

Mark Owyong, Jonathan Chou, Renske van den Bijgaart, Niwen Kong, Gizem Efe, Carrie Maynard, Dalit Talmi-Frank, Inna Solomonov, Charlotte Koopman, Elin Hadler-Olsen, Mark Headley, Charlene Lin, Chih-Yang Wang, Irit Sagi, Zena Werb, and Vicki Plaks

DOI: <https://doi.org/10.26508/lsa.201800226>

Corresponding author(s): Vicki Plaks, UCSF and Zena Werb, University of California

Review Timeline:

Submission Date:	2018-10-26
Editorial Decision:	2018-10-29
Revision Received:	2019-01-28
Editorial Decision:	2019-02-11
Revision Received:	2019-09-25
Editorial Decision:	2019-10-02
Revision Received:	2019-10-09
Accepted:	2019-10-09

Scientific Editor: Andrea Leibfried

Transaction Report:

Please note that the manuscript was previously reviewed at another journal and the reports were taken into account in the decision-making process at Life Science Alliance. Since the original reviews are not subject to Life Science Alliance's transparent review process policy, the reports and author response cannot be published.

October 29, 2018

Re: Life Science Alliance manuscript #LSA-2018-00226-T

Dr. Vicki Plaks
UCSF
Anatomy
513 Parnassus Ave
HSW 1301
San Francisco, CA 94143

Dear Dr. Plaks,

Thank you for transferring your manuscript entitled "Matrix Metalloproteinase 9 Modulates the Metastatic Cascade and Immune Landscape for Breast Cancer Anti-Metastatic Therapy" to Life Science Alliance. The manuscript was assessed by expert reviewers at another journal before, and the editors transferred those reports to us with your permission.

The reviewers noted that your work adds to our understanding of the early process of metastatic colonisation in the lungs. However, they also think that the conclusions put forward are not sufficiently supported given the low number of mice examined, and they were concerned by the statistical analysis performed. You already provided a preliminary point-by-point response upfront, outlining how you could address the reviewers' concerns. Based on this outline we would like to invite you to submit a revised version of your manuscript for publication in Life Science Alliance, addressing the reviewers' concerns as proposed. We think it would be particularly important to experimentally address the concerns # 2, 3, 4, and 7 of reviewer #2, all pertinent to the major criticisms mentioned above.

The typical timeframe for revisions is three months. Please note that papers are generally considered through only one revision cycle.

Thank you for this interesting contribution to Life Science Alliance. We are looking forward to receiving your revised manuscript.

Sincerely,

- A letter addressing the reviewers' comments point by point.
- An editable version of the final text (.DOC or .DOCX) is needed for copyediting (no PDFs).
- High-resolution figure, supplementary figure and video files uploaded as individual files: See our detailed guidelines for preparing your production-ready images, <http://life-science-alliance.org/authorguide>
- Summary blurb (enter in submission system): A short text summarizing in a single sentence the study (max. 200 characters including spaces). This text is used in conjunction with the titles of papers, hence should be informative and complementary to the title and running title. It should describe the context and significance of the findings for a general readership; it should be written in the present tense and refer to the work in the third person. Author names should not be mentioned.

B. MANUSCRIPT ORGANIZATION AND FORMATTING:

Full guidelines are available on our Instructions for Authors page, <http://life-science-alliance.org/authorguide>

January 28, 2019

Dr. Andrea Leibfried
Executive Editor
Life Science Alliance

Re: LSA-2018-00226-T Revised Manuscript

Dear Dr. Andrea Leibfried,

Below are the changes we have made in our revised manuscript per what was requested for the transfer.

1. Statistical analyses have been updated for all figures to be mathematically correct dependent on the sample size. Unless otherwise noted in the figure legend, all statistical analysis performed is a two-tailed student's t-test with error bars depicting standard error.
2. Per the experimental metastasis mouse experiments in the double transgenic PyMT/MMP9 mouse model (figures 2E and 2F), we made significant attempts to run this experiment but ran into an issue with generating enough mice within the time allotted for this revision. These mice have been more difficult than assumed to generate a sufficient cohort size due to the low odds of receiving a female MMP9^{KO/KO} mouse that is also MMTV-PyMT⁺. While we have expanded our colony significantly, we were not able to generate enough mice to rerun the variety of combinations of double transgenic mouse experiments that we had proposed. However, we believe that increasing the number of MMTV-PyMT⁺/MMP9^{WT} mice will not aid in the conclusions made in figures 2E and 2F since we have shown in other experiments that MMTV-PyMT⁺ mice are more susceptible to metastatic seeding and that the MMTV-PyMT⁺/MMP9^{KO/KO} mice have reduced susceptibility with a sample size of 2 and low variance between the animals. Since animal-to-animal variance was a major concern with the MMTV-PyMT mouse model, we believe that our small margin of variation shown in figure 2F is sufficient to support the claims we make. With that said, we would like to reiterate that this experiment is a proof-of-concept for the relevance of MMP9 in this model and set the stage for our more conclusive pharmacological experiments. These feature our monoclonal antibody against active MMP9 and its interrogation for therapeutic purposes. We believe that the focus should not be on a preventative approach, which is the MMTV-PyMT⁺/MMP9^{KO/KO} mice, since the clinical relevance for utilizing this

type of inhibitor against gelatinases exist in a therapeutic setting. We added this clarification to the manuscript.

3. We have increased our 'n' in figure 2G and addressed the statistical analysis comment mentioned by reviewer number two. Due to increasing our sample size, we can now justifiably run a standard error analysis.
4. For the 'n' in figures 1E, 3, and supplemental figure 3, we have added in the number of mice used for each experiment. This was an oversight that we did not catch prior to our initial submission and is now corrected.
5. For the low 'n' mentioned by Reviewer #2 for supplementary figure 2F, we have increased our 'n' for the age range(s) that were feasible given the timeline and the number of double transgenic mice we were able to generate. As mentioned earlier in point 2 above, we ran into some unexpected difficulties in generating large quantities of these mice, in particular, aging them out to 4 months and beyond. Thus, once again, we would like to touch on the basis that this experiment is non-critical to our overall understanding of how inhibition of MMP9 can alter early metastatic growth. This experiment is used to show that in a preventative setting, which is not particularly clinically relevant, MMP9 can delay the onset of metastatic colonization. This panel supports claims we make when using our therapeutic monoclonal antibody, which delays the onset of metastatic growth in the early stages of dissemination. For MMTV-PyMT⁺ mice this is particularly relevant for females around ages 12-14 weeks, the dataset we chose to focus on and strengthen.
6. Our figure legend and text for figure 3 has been updated to align with the message we are conveying.
7. We have clarified the analytical method used within our scratch assay (figure 5B) to allow the reader(s) to fully understand what the lines and graphs mean in our figure. We have also added details in our quantification of immunofluorescence for CD8a quantification (figure 6H) to allow the reader(s) to understand the region of interest we used for each image to get to the conclusions we have reached.
8. We have added text in our results regarding our double transgenic mouse model experiment being utilized as supporting evidence and proof-of-concept for our pharmacological approach to the importance of MMP9 in early metastatic growth. We have also added commentary in the discussion on where we envision utilizing an anti-MMP9 therapeutic regimen and how it would add to currently on-going clinical trials that often focus on late-stage tumors.

February 11, 2019

Re: Life Science Alliance manuscript #LSA-2018-00226-TR

Dr. Vicki Plaks
UCSF
Anatomy
513 Parnassus Ave
HSW 1301
San Francisco, CA 94143

Dear Dr. Plaks,

Thank you for submitting your revised manuscript entitled "MMP9 Modulates the Metastatic Cascade and Immune Landscape for Breast Cancer Anti-Metastatic Therapy" to Life Science Alliance. We had asked you to revise your work in response to the reviewer concerns raised at another journal. Specifically, we asked you to address points 2, 3, 4, and 7 of reviewer #2, and you had provided a preliminary point-by-point response to outline how you would do so.

We appreciate the changes included in the revised version and we realize that you have have addressed point 3 and 7 of reviewer #2. The revised version does not address point 4 as previously proposed. More importantly however, point 2 hasn't been addressed due to problems to generate more mice. We appreciate your explanation, but as addressing this point was a revision requirement, we are sorry to say that we concluded that we cannot offer publication here. This view was also supported by a breast cancer expert additionally consulted on your work who stated: "It is problematic reporting data based on n=1 in general and here in particular given variability in the PyMT model and even if it is not the main line of the article. I would therefore not recommend publication."

We appreciate the effort that has gone into the revisions and regret that the outcome is not more positive.

Kind regards,
Andrea

30 September, 2019

Dear Dr. Andrea Leibfried,

We have already provided a point-by-point response to the previous concerns that arose from the three reviewers. We have added comments that pertain specifically to the major points that you had mentioned in correspondence with us via email which focus on increasing the 'n' for certain datasets to enable publication. Since now all the requests are complete, we truly hope this will help expedite the publication of our manuscript in Life Science Alliance.

For ease of review, we will start with the correspondence where you ask us to complete a set of experiments for further consideration, all of which we have completed.

- **Addressing post review requests from Dr. Andrea Leibfried:**

1. We have detailed the number of mice used for our experiments in **Figure 3** within the legend.
2. We have increased the number of mice used in **Figure 6H** from n=1 isotype to n=5 isotype and n=1 SDS3 to n=5 SDS3 treated mouse lungs. Each animal had 3 serial sections analyzed to account for depth variability.
3. We have increased the number of mice used in **Figure 2E** and **Figure 2F** from n=1 MMTV-PyMT+/MMP9 WT to n=4 and from n=2 MMTV-PyMT+/MMP9 KO to n=3. Each animal had 3 serial sections analyzed to account for depth variability.
4. We have removed supplementary mouse dataset S2D-S2H as previously agreed upon per our discussion. We mutually agreed to remove this supplementary piece of data since our focus in this work is on early metastatic dissemination and growth rather than on late stage processes involving gelatinases. Specifically, this work is centered on how MMP9 plays a pivotal role in early phases of metastatic dissemination to facilitate metastatic niche formation and increase susceptibility to seeding by circulating tumor cells. On the contrary, at the later stages of metastatic tumors, similar as to what was previously observed in the clinic, it becomes increasingly difficult to find a therapeutic agent that targets MMPs and can slow the progression of a fully established lesion. These lesions are potentially self-sufficient to survive without the necessity of MMPs supplied by the metastatic niche.
5. We updated the references with *Juric et al PLoS One 2018*. This manuscript came out in the past year and showed MMP9 enacting anti-tumor immunity through T cell trafficking in an orthotopic model of HER2 breast cancer. This further reiterates the important role that MMP9 is playing in promoting a pro-inflammatory tumor microenvironment.

October 2, 2019

RE: Life Science Alliance Manuscript #LSA-2018-00226-TRR-A

Dr. Vicki Plaks
UCSF
Anatomy
513 Parnassus Ave
HSW 1301
San Francisco, CA 94143

Dear Dr. Plaks,

Thank you for submitting your revised manuscript entitled "MMP9 Modulates the Metastatic Cascade and Immune Landscape for Breast Cancer Anti-Metastatic Therapy". We appreciate the introduced changes and, importantly, the increase in mice used for the analyses, and we would thus be happy to publish your paper in Life Science Alliance pending final revisions necessary to meet our formatting guidelines:

- please note that we display suppl figures in-line in the HTML version of the paper. These cannot run over several pages, please adapt Fig S6 accordingly (please consider making two figures out of this figure)
- please add legends to your manuscript text for the movies provided, the movies will also be displayed in-line in the HTML version of the paper
- please mention the arrows depicted in figure panels 1E, 2B, 6H in the figure legends
- please link your profile in our submission system to your ORCID iD, you should have received an email with instructions on how to do so; all corresponding authors should do this, please

A. FINAL FILES:

-- High-resolution figure, supplementary figure and video files uploaded as individual files: See our

detailed guidelines for preparing your production-ready images, <http://www.life-science-alliance.org/authors>

B. MANUSCRIPT ORGANIZATION AND FORMATTING:

Sincerely,

Andrea Leibfried, PhD
Executive Editor
Life Science Alliance
Meyerhofstr. 1
69117 Heidelberg, Germany
t +49 6221 8891 502

e.a.leibfried@life-science-alliance.org
www.life-science-alliance.org

October 9, 2019

RE: Life Science Alliance Manuscript #LSA-2018-00226-TRRR

Dr. Vicki Plaks
UCSF
Anatomy
513 Parnassus Ave
HSW 1301
San Francisco, CA 94143

Dear Dr. Plaks,

Thank you for submitting your Research Article entitled "MMP9 Modulates the Metastatic Cascade and Immune Landscape for Breast Cancer Anti-Metastatic Therapy". It is a pleasure to let you know that your manuscript is now accepted for publication in Life Science Alliance. Congratulations on this interesting work.

DISTRIBUTION OF MATERIALS:

Again, congratulations on a very nice paper. I hope you found the review process to be constructive and are pleased with how the manuscript was handled editorially. We look forward to future exciting submissions from your lab.

Sincerely,
